# Transport Score Climbing: Variational Inference Using Forward KL and Adaptive Neural Transport

## Abstract

Variational inference often minimizes the "reverse" Kullbeck-Leibler (KL) $D_{KL}(q||p)$ from the approximate distribution $q$ to the posterior $p$. Recent work studies the "forward" KL $D_{KL}(p||q)$, which unlike reverse KL does not lead to variational approximations that underestimate uncertainty. Markov chain Monte Carlo (MCMC) methods were used to evaluate the expectation in computing the forward KL. This paper introduces Transport Score Climbing (TSC), a method that optimizes $D_{KL}(p||q)$ by using Hamiltonian Monte Carlo (HMC) but running the HMC chain on a transformed, or warped, space. A function called the transport map performs the transformation by acting as a change-of-variable from the latent variable space. TSC uses HMC samples to dynamically train the transport map while optimizing $D_{KL}(p||q)$. TSC leverages synergies, where better transport maps lead to better HMC sampling, which then leads to better transport maps. We demonstrate TSC on synthetic and real data, including using TSC to train variational auto-encoders. We find that TSC achieves competitive performance on the experiments.

## 1 Introduction

A main goal in probabilistic modeling and inference is to find the posterior distribution of latent variables given observed data (Gelman et al., 2013). Probabilistic modeling allows using both structured knowledge and flexible parameterizations, including neural networks, but the posterior is often intractable. In this situation, we can resort to approximate inference to estimate the posterior distribution (Bishop, 2006).

Variational Inference (VI) is an optimization-based approximate inference method. It posits a family of distributions, and chooses a distribution $q$ in that family to approximate the posterior $p$ of a probabilistic model. It is a popular method for complex models because of its computational convenience, particularly when optimizing the "reverse", or "exclusive", Kullbeck-Leibler (KL) divergence $D_{KL}(q||p)$ through stochastic gradient descent (SGD) (Jordan et al., 1999; Hoffman et al., 2013; Blei et al., 2017).

However, reverse VI - VI that uses the reverse KL - leads to approximations that may underestimate the uncertainty in $p$ (Minka, 2005; Yao et al., 2018). As an alternative, forward VI minimizes the "forward", or "inclusive", KL $D_{KL}(p||q)$. This approach better captures posterior uncertainty, but it is more computationally challenging (Bornschein & Bengio, 2015; Gu et al., 2015; Finke & Thiery, 2019; Naesseth et al., 2020).

Another approach to approximate inference is Markov chain Monte Carlo (MCMC). MCMC methods sample from a Markov chain whose stationary distribution is the posterior, and produce good samples if run for long enough. However, in practice MCMC methods can be more computationally demanding than reverse VI in that they can take many iterations to converge.

To combine the advantages of both paradigms, Naesseth et al. (2020) introduce Markovian score climbing (MSC). MSC is a variational method for minimizing the forward $D_{KL}(p||q)$, which uses a Markov chain to approximate its intractable expectation over $p$. MSC uses an MCMC chain to approximate the expectation without asymptotic biases. However, this method uses basic MCMC kernels that can lead to slow exploration of the sampling space.

In this paper, we develop transport score climbing (TSC), a new algorithm that reliably and efficiently minimizes $D_{KL}(p||q)$. TSC uses the MSC framework, but replaces the simple MCMC kernel with a Hamiltonian Monte Carlo (HMC) on a transformed, or warped, space (Marzouk et al., 2016; Mangoubi & Smith, 2017; Hoffman et al., 2019). In particular, we adaptively transform the HMC sampling space, where the transformation is based on the current iteration of the variational approximation.

In more detail, TSC optimizes a normalizing flow (Rezende & Mohamed, 2015), where the flow (or, equivalently, transport map) is trained from HMC samples from the warped space. Thus, TSC trains its transport map from scratch and leverages a synergy between the Markov chain and the variational approximation: an updated transport map improves the HMC trajectory, and the better HMC samples help train the transport map.

Finally, we show how TSC is amenable to SGD on large-scale IID data. To this end, we use TSC to improve training of deep generative models with a variational autoencoder (VAE) (Kingma & Welling, 2014; Rezende et al., 2014).

**Contributions.** 1) We minimize $D_{KL}(p||q)$ with flow posteriors and an adaptive HMC kernel. The HMC kernel reuses the flow posterior to warp the underlying latent space for more efficient sampling. 2) Under the framework of VI with $D_{KL}(p||q)$, we show that the transport map of the warped space can be trained adaptively, instead of requiring a separate pre-training suggested by previous methods. 3) Empirical studies show that TSC more closely approximates the posterior distribution than both reverse VI and MSC. Furthermore, we use this methodology to develop a novel VAE algorithm competitive against four benchmarks. TSC continuously runs HMC chains and requires no reinitializations from the variational approximation $q$ at each epoch, but these reinitializations are used by previous methods.

**Related Work.** Forward VI is explored by several approaches. Bornschein & Bengio (2015); Finke & Thiery (2019); Jerfel et al. (2021) study VI with $D_{KL}(p||q)$ by using importance sampling (IS), and Gu et al. (2015) uses sequential Monte Carlo (SMC). Dieng & Paisley (2019) combines IS and VI in an expectation maximization algorithm. IS and SMC introduce a non-vanishing bias that leads to a solution which optimizes $D_{KL}(p||q)$ and the marginal likelihood only approximately (Naesseth et al., 2019; 2020). Closest to the method proposed here is Naesseth et al. (2020); Ou & Song (2020); Gabrié et al. (2021), which all use MCMC kernels to minimize $D_{KL}(p||q)$. Ou & Song (2020); Gabrié et al. (2021) can be considered to be instances of MSC (Naesseth et al., 2020). We build on MSC and propose to use the more robust HMC kernel together with a space transformation. The work of Kim et al. (2022), running parallel Markov chains for improved performance, can be combined with TSC for potential further gains.

Mangoubi & Smith (2017) show that MCMC algorithms are more efficient on simpler spaces, such as on strongly log-concave targets. Marzouk et al. (2016); Hoffman et al. (2019) use transformations to create warped spaces that are easy to sample from. The transformation is defined by functions called "transport maps" that are pre-trained by reverse VI. The proposed algorithm differs in the optimization objective and by learning the transport map together with model parameters end-to-end.

Using MCMC to learn model parameters based on the maximum marginal likelihood is studied in many papers, e.g., Gu & Kong (1998); Kuhn & Lavielle (2004); Andrieu & Moulines (2006); Andrieu & Vihola

(2014). In contrast TSC proposes a new method for the same objective, by adapting the MCMC kernel using VI.

Kingma & Welling (2014); Rezende et al. (2014) introduce variational autoencoders (VAE) where both the generative model and the approximate posterior are parameterized by neural networks. They optimize a lower bound to the marginal log-likelihood, called the evidence lower bound (ELBO), with the reparameterization trick. Salimans et al. (2015); Caterini et al. (2018) incorporate MCMC or HMC steps to train a modified ELBO with lower variance. Hoffman (2017); Hoffman et al. (2019); Ruiz et al. (2021) instead formulate the optimization as maximum likelihood while utilizing MCMC methods. The work proposed here also targets maximum likelihood, but we neither augment the latent variable space (Ruiz et al., 2021) nor reinitialize the Markov kernel from the posterior at each epoch (Hoffman, 2017; Hoffman et al., 2019). Instead, we continuously run the Markov kernel on the warped latent variable space.

## 2 Background

Let $p(\mathbf{x}, \mathbf{z})$ be a probabilistic model, with $\mathbf{z}$ as latent variables and $\mathbf{x}$ as observed data. The probabilistic model factorizes into the product of the likelihood $p(\mathbf{x}|\mathbf{z})$ and prior $p(\mathbf{z})$, which are known and part of the modeling assumptions. A main goal of Bayesian inference is to calculate or approximate the posterior distribution of latent variables given data, $p(\mathbf{z}|\mathbf{x})$.

VI approximates the posterior by positing a family of distributions $\mathcal{Q}$, where each distribution takes the form $q(\mathbf{z}; \lambda)$ with variational parameters $\lambda$. The most common approach, reverse VI, minimizes the reverse KL using gradient-based methods: $\min_\lambda D_{KL}(q(\mathbf{z}; \lambda)||p(\mathbf{z}|\mathbf{x}))$. The main strength of reverse VI is computational convenience.

### 2.1 Variational Inference with Forward KL

Reverse VI often underestimates the uncertainty in $p$. An alternative approach, which is the focus of this work, is to minimize the forward KL: $\min_\lambda D_{KL}(p(\mathbf{z}|\mathbf{x})||q(\mathbf{z}; \lambda))$. While more challenging to work with, this objective does not lead to approximations that underestimate uncertainty (Naesseth et al., 2020). Moreover, if $\mathcal{Q}$ is the subset of exponential family distributions with sufficient statistics $T$ and $\mathbb{E}_p[T]$ exist and are finite, the optimal $q$ matches the expected sufficient statistics values under the posterior $p$ exactly.

The forward KL divergence from $p$ to $q$ is

$$D_{KL}(p(\mathbf{z}|\mathbf{x})||q(\mathbf{z}; \lambda)) := \mathbb{E}_{p(\mathbf{z}|\mathbf{x})}\left[\log \frac{p(\mathbf{z}|\mathbf{x})}{q(\mathbf{z}; \lambda)}\right]. \tag{1}$$

To minimize eq. (1), the gradient w.r.t. the variational parameters is,

$$\mathbb{E}_{p(\mathbf{z}|\mathbf{x})}[-\nabla_\lambda \log q(\mathbf{z}; \lambda)]. \tag{2}$$

Approximating the expectation over the unknown posterior $p(\mathbf{z}|\mathbf{x})$ is a major challenge. Bornschein & Bengio (2015); Gu et al. (2015) approximate the expectation in eq. (2) through importance sampling and sequential Monte Carlo, but these methods gives estimates of the gradient with systematic bias.

In this work we leverage Markovian score climbing (MSC) (Naesseth et al., 2020), which uses samples $\mathbf{z}$ from an MCMC kernel with the posterior $p(\mathbf{z}|\mathbf{x})$ as its stationary distribution. The resulting SGD method leads to an algorithm that provably minimizes $D_{KL}(p||q)$ (Naesseth et al., 2020).

**Normalizing Flow.** In this work we focus on the variational family of normalizing flows. Normalizing flows transform variables with simple distributions and build expressive approximate posteriors (Rezende & Mohamed, 2015; Tabak & Turner, 2013), and are tightly linked with warped space HMC. Given a $d$-dimensional latent variable $\mathbf{z}$, the transformation uses an invertible, smooth, trainable function $T_\lambda : \mathbb{R}^d \mapsto \mathbb{R}^d$ and introduces a random variable $\boldsymbol{\epsilon}$ with a simple distribution $q_0(\boldsymbol{\epsilon})$, oftentimes an isotropic Gaussian. Using the change-of-variable identity, the probability density function $q$ of $T_\lambda(\boldsymbol{\epsilon})$ is,

$$q(T_\lambda(\boldsymbol{\epsilon})) = q_0(\boldsymbol{\epsilon})\Big|\det\frac{dT_\lambda}{d\boldsymbol{\epsilon}}\Big|^{-1},$$

where $\frac{dT_\lambda}{d\boldsymbol{\epsilon}}$ is the Jacobian matrix.

## 2.2 Hamiltonian Monte Carlo (HMC) and Neural Transport HMC

The HMC kernel used in the algorithm proposed below is closely related to Neural Transport HMC (NeutraHMC), proposed by Hoffman et al. (2019). NeutraHMC simplifies the geometry of the sampling space through neural network-parameterized transport maps. Compared to HMC, it explores the target distribution more efficiently. We briefly explain HMC and NeutraHMC.

**Hamiltonian Monte Carlo.** HMC is an MCMC algorithm that produces larger moves in latent variable $\mathbf{z}$ by introducing "momentum" variables $\mathbf{m}$ of the same dimension as $\mathbf{z}$ (Duane et al., 1987; Neal, 2011). It constructs a joint proposal on the augmented space $(\mathbf{z}, \mathbf{m})$ to target $p(\mathbf{z}|\mathbf{x})p(\mathbf{m})$, where $\mathbf{x}$ is data. A common choice for the distribution $p(\mathbf{m})$ is $N(0, I)$.

In a given iteration, a proposal involves $L$ "leapfrog steps" of step-size $s$, where the $l$-th leapfrog step is defined by

$$\mathbf{m}^{(l)'} = \mathbf{m}^{(l-1)} + \frac{1}{2}s\frac{d\log p(\mathbf{x}, \mathbf{z}^{(l)})}{d\mathbf{z}^{(l)}},$$
$$\mathbf{z}^{(l)} = \mathbf{z}^{(l-1)} + s\mathbf{m}^{(l)'},$$
$$\mathbf{m}^{(l)} = \mathbf{m}^{(l)'} + \frac{1}{2}s\frac{d\log p(\mathbf{x}, \mathbf{z}^{(l)'})}{d\mathbf{z}^{(l)'}},$$

starting from $(\mathbf{z}^{(0)}, \mathbf{m}^{(0)})$, $\mathbf{m}^{(0)} \sim N(0, I)$, $\mathbf{z}^{(0)}$ randomly initialized or set to the previous MCMC state. The final leapfrog step gives the proposed state $(\mathbf{z}^{(L)}, \mathbf{m}^{(L)})$. The new state is accepted with probability $\min\{1, \frac{p(\mathbf{x},\mathbf{z}^{(L)})p(\mathbf{m}^{(L)})}{p(\mathbf{x},\mathbf{z})p(\mathbf{m})}\}$ (Neal, 2011; Robert & Casella, 2004).

**HMC on Warped Space.** Marzouk et al. (2016); Mangoubi & Smith (2017); Hoffman et al. (2019) propose running MCMC methods on a simpler geometry through transforming, or warping, the sampling space with a transport map. A transport map is defined as a parameterized function $T_\lambda(\cdot)$. The warped space is defined by the change in variable $\boldsymbol{z_0} = T_\lambda^{-1}(\mathbf{z})$ for $\mathbf{z} \sim p(\mathbf{z}|\mathbf{x})$. If $T_\lambda$ is chosen well, $\boldsymbol{z_0}$ will be simpler than $\mathbf{z}$ to sample. The target distribution in the MCMC algorithm is defined as the distribution of $\boldsymbol{z_0}$. Each $\boldsymbol{z_0}^{(k)}$ generated by MCMC at the $k$-th iteration is passed to the transport map with $\mathbf{z}^{(k)} = T_\lambda(\boldsymbol{z_0}^{(k)})$. $(\mathbf{z}^{(1)}, \mathbf{z}^{(2)}, ...)$ then have the true target distribution as its stationary distribution, but with faster mixing than MCMC on the original space. Hoffman et al. (2019) introduces NeutraHMC that uses HMC instead of general MCMC. NeutraHMC utilizes both affine and neural network transport maps that are pretrained using VI based on $KL(q||p)$.

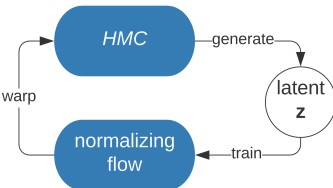

Figure 1: Outline of the TSC algorithm. HMC generates samples of latent variable $\mathbf{z}$ that are used to train the normalizing flow transformation. The refined transformation further improves the geometry of HMC, which goes on to generate the next sample.

## 3 Transport Score Climbing

We now develop Transport Score Climbing (TSC), a method for VI with forward $D_{KL}(p||q)$. TSC uses HMC on warped space to estimate the intractable expectation in the gradient (eq.(2)). A transport map is defined to act as both flow transformation in the variational posterior $q$ and mapping between HMC sampling spaces. As $q$ is updated, the mapping is updated simultaneously which further refines the HMC sampling space. Figure 1 shows the synergy between HMC sampling and variational approximation. Under conditions (Section A.1), TSC converges to a local optimum of $D_{KL}(p||q)$.

### 3.1 Types of Transport Maps

Let $\boldsymbol{\epsilon} \sim N(0, I_d)$, let $T_\lambda(\cdot)$ be a function $\mathbb{R}^d \mapsto \mathbb{R}^d$, or transport map, with trainable parameter $\lambda$, and define the variational distribution $q(\mathbf{z}; \lambda)$ such that $\mathbf{z} = T_\lambda(\boldsymbol{\epsilon}) \sim q(\mathbf{z}; \lambda)$. The variational distribution and transport map share trainable parameters. We consider three concrete examples.

**Affine Transformation.** Consider an affine transformation $T_\lambda(\boldsymbol{\epsilon}) = \boldsymbol{\mu} + \boldsymbol{\sigma} \odot \boldsymbol{\epsilon}$, where $\odot$ denotes elementwise multiplication. The variational distribution is $q(\mathbf{z}; \lambda) = N(\mathbf{z}; \boldsymbol{\mu}, \boldsymbol{\sigma}^2 I)$. In the empirical studies, Section 4, we find that the affine transport map is simple and effective.

**IAF Transformation.** A popular flow transformation is the inverse autoregressive flow (IAF) (Kingma et al., 2016). $T_\lambda$ is chosen with the autoregressive property, that is, along each dimension $i$ of $\boldsymbol{\epsilon}$,

$$T_i(\boldsymbol{\epsilon}) = \boldsymbol{\epsilon}_i \sigma_i(\boldsymbol{\epsilon}_{1:i-1}; \phi) + \mu_i(\boldsymbol{\epsilon}_{1:i-1}; \phi).$$

Here $\mu$ and $\sigma$ are neural networks that act as shift and scale functions. IAF is flexible because of neural networks; its determinant is cheap to compute because of the autoregressive property that leads to a lower triangular Jacobian matrix. However, the inverse IAF $T_\lambda^{-1}(\mathbf{z})$, required to evaluate the density $q$, is costly to compute. Thus, we only use IAF in studies where latent variables are low-dimensional.

**RealNVP Transformation.** RealNVP is an alternative to IAF with slightly less flexibility for the same number of parameters but fast invertibility (Dinh et al., 2016). RealNVP uses an affine coupling layer to transform the input $\boldsymbol{\epsilon}$. In practice, we use a checkerboard binary mask $b$ to implement the transformation, as detailed in Dinh et al. (2016),

$$T(\boldsymbol{\epsilon}) = b \odot \boldsymbol{\epsilon} + (1 - b) \odot \Big( \boldsymbol{\epsilon} \odot \exp\big(\sigma(b \odot \boldsymbol{\epsilon})\big) + \mu(b \odot \boldsymbol{\epsilon})\Big).$$

where $\mu$ and $\sigma$ are also neural networks. The idea is that the part of $\boldsymbol{\epsilon}$ that is transformed by neural networks depend only on the other part of $\boldsymbol{\epsilon}$ that goes through the identity function. This construction allows for fast inversion.

---

**Algorithm 1** TSC

---

**Input:** Probabilistic model $p(\mathbf{z}, \mathbf{x}; \theta)$; transformation $T_\lambda : \mathbb{R}^d \mapsto \mathbb{R}^d$; HMC kernel $H[\mathbf{z_0}^{(k+1)}|\mathbf{z_0}^{(k)}; \lambda, \theta]$ with target distribution $p(\mathbf{z_0}|\mathbf{x}; \theta, \lambda)$ and initial state $\mathbf{z_0}^{(0)}$; variational distribution $q(\mathbf{z}; \lambda)$; step-sizes $\alpha_1, \alpha_2$. $\lambda, \theta$ randomly initialized.

**Output:** $\lambda, \theta$.

   **for** $k \in \{0, 1, 2, ...\}$ **do**
      $\mathbf{z_0}^{(k+1)} \sim H[\mathbf{z_0}^{(k+1)}|\mathbf{z_0}^{(k)}; \lambda, \theta]$.
      $\mathbf{z} = T_\lambda(\mathbf{z_0}^{(k+1)})$.
      $\mathbf{z} = \text{stop-gradient}(\mathbf{z})$.
      $\lambda = \lambda - \alpha_1 \nabla_\lambda[-\log q(\mathbf{z}; \lambda)]$.
      $\theta = \theta - \alpha_2 \nabla_\theta[-\log p(\mathbf{x}|\mathbf{z}; \theta) - \log p(\mathbf{z})]$.
      $\mathbf{z_0}^{(k+1)} = T_\lambda^{-1}(\mathbf{z})$.
   **end for**

---

Both IAF and RealNVP flow transformations can be stacked to form more expressive approximations. Let $T_\lambda^{(l)}$ denote one IAF or RealNVP transformation. We stack L transformations, and define the transport map as $T_\lambda(\boldsymbol{\epsilon}) = T_{\lambda_L}^{(L)} \circ ... \circ T_{\lambda_1}^{(1)}(\boldsymbol{\epsilon})$. The variational distribution $q(\mathbf{z}; \lambda)$ is a flow-based posterior with $q(\mathbf{z}; \lambda) = N(\boldsymbol{\epsilon}; 0, I) \left| \det \frac{dT_\lambda(\boldsymbol{\epsilon})}{d\boldsymbol{\epsilon}} \right|^{-1}$.

### 3.2 VI with HMC on Warped Space

In order to sample the latent variables $\mathbf{z}$, we define the target of the HMC kernel $H(\cdot|\mathbf{z_0})$ as the distribution of $\mathbf{z_0} = T_\lambda^{-1}(\mathbf{z})$, $\mathbf{z} \sim p(\mathbf{z}|\mathbf{x})$,

$$p(\mathbf{z_0}|\mathbf{x}; \lambda) \propto p(\mathbf{x}, T_\lambda(\mathbf{z_0}))|\det J_{T_\lambda}(\mathbf{z_0})|, \tag{3}$$

where $J_f(x)$ is the Jacobian matrix of function $f$ evaluated at $x$. This means that we are sampling on the warped space defined by $T_\lambda(\cdot)$ rather than the original space of latent variables. After $\mathbf{z_0}$ is sampled, we pass it to the transport map with $\mathbf{z} = T_\lambda(\mathbf{z_0})$ to acquire the latent variable sample. As in MSC (Naesseth et al., 2020), we do not re-initialize the Markov chain at each iteration, but use the previous sample $\mathbf{z}^{(k)}$ to both estimate the gradient and serve as the current state of the HMC kernel $H$ to sample $\mathbf{z}^{(k+1)}$.

A crucial element is that the transport map $T_\lambda$ is trained jointly as we update the $D_{KL}(p||q)$ objective in eq.(2). This is because the map is also the flow transformation part of the variational distribution $q$. Specifically, HMC at iteration $k$ uses variational parameters of the previous iteration, $\lambda^{(k-1)}$, in its target distribution (eq.(3)) at iteration $k$. By construction, if $q$ is close to the true posterior, target $p(\mathbf{z_0}|\mathbf{x}; \lambda)$ will be close to the isotropic Gaussian. Therefore, TSC keeps refining the geometry of the HMC sampling space throughout the training process.

#### 3.2.1 Model Parameters

The probabilistic model $p(\mathbf{z}, \mathbf{x}; \theta)$ can also contain unknown parameters $\theta$. The corresponding warped space posterior is

$$p(\mathbf{z_0}|\mathbf{x}; \lambda, \theta) \propto p(\mathbf{x}, T_\lambda(\mathbf{z_0}); \theta)|\det J_{T_\lambda}(\mathbf{z_0})|. \tag{4}$$

Taking samples from the true posterior $p(\mathbf{z}|\mathbf{x}; \theta)$ allows one to learn $\theta$ using maximum likelihood, optimizing the marginal likelihood $p(\mathbf{x}; \theta)$. This fact follows from the Fisher identity, which writes the gradient of the

marginal likelihood as an expectation over the posterior,

$$
\begin{aligned}
\nabla_\theta \log p(\mathbf{x}; \theta) &= \nabla_\theta \log \int p(\mathbf{z}, \mathbf{x}; \theta) d\mathbf{z} \\
&= \mathbb{E}_{p(\mathbf{z}|\mathbf{x};\theta)}[\nabla_\theta \log p(\mathbf{z}, \mathbf{x}; \theta)].
\end{aligned}
\tag{5}
$$

The expectation above is estimated by the same HMC sample $\mathbf{z}^{(k)}$ that is used to update variational parameters $\lambda$. Additionally, the HMC kernel at iteration $k$ uses model parameters of the previous iteration, $\theta^{(k-1)}$, in its target distribution (eq.(4)) at iteration $k$. Under conditions on the step size, Markov chain, and gradients, the corresponding algorithm can be shown to optimize the log-marginal likelihood and converges to a local optima by application of (Gu & Kong, 1998, Theorem 1), see Section A.1 for details.

Algorithm 1 summarizes TSC for learning $\lambda$ and $\theta$.

### 3.3 Amortized Inference

When the dataset $\mathbf{x} = (\mathbf{x}_1, ..., \mathbf{x}_n)$ is i.i.d. with empirical distribution $\widehat{p}(\mathbf{x})$ each $\mathbf{x}_i$ has its own latent variable $\mathbf{z}$. Amortized inference then uses the approximate posterior $q(\mathbf{z}|\mathbf{x}; \lambda)$ instead of a separate $q(\mathbf{z}; \lambda_i)$ for each $\mathbf{x}_i$. In amortized inference, variational parameters $\lambda$ are shared across data-points $\mathbf{x}_i$. It is known as a VAE when both the likelihood $p(\mathbf{x}|\mathbf{z}; \theta)$ and the approximate posterior $q$ are parameterized by neural networks.

TSC conducts maximum likelihood and VI with $D_{KL}(p||q)$ on VAE and is amenable to SGD with mini-batches. Following derivations from Naesseth et al. (2020), the gradient with respect to $\lambda$ is

$$
\mathbb{E}_{\widehat{p}(\mathbf{x})}\big[\nabla_\lambda \mathrm{KL}(p(\mathbf{z}|\mathbf{x};\theta)||q(\mathbf{z}|\mathbf{x};\lambda))\big] \approx \frac{1}{M} \sum_{i=1}^{M} \mathbb{E}_{p(\mathbf{z}|\mathbf{x}_i)}[-\nabla_\lambda \log q(\mathbf{z}|\mathbf{x}_i; \lambda)],
\tag{6}
$$

where $M$ is the mini-batch size. For model learning, we similarly estimate the gradient using eq. (5),

$$
\begin{aligned}
\mathbb{E}_{\widehat{p}(\mathbf{x})}\big[\nabla_\theta \log p(\mathbf{x}; \theta)\big] &= \mathbb{E}_{\widehat{p}(\mathbf{x})}\big[\mathbb{E}_{p(\mathbf{z}|\mathbf{x};\theta)}[\nabla_\theta[\log p(\mathbf{x}|\mathbf{z}; \theta) + \log p(\mathbf{z})]]\big] \\
&\approx \frac{1}{M} \sum_{i=1}^{M} \mathbb{E}_{p(\mathbf{z}|\mathbf{x}_i;\theta)}[\nabla_\theta[\log p(\mathbf{x}_i|\mathbf{z}; \theta) + \log p(\mathbf{z})]].
\end{aligned}
\tag{7}
$$

The expectations are approximated using HMC samples, as in Algorithm 1. Similarly with the non-amortized case, we do not re-initialize the Markov chain at each iteration, but approximate the expectation by running one step of the Markov chain on the previous sample $\mathbf{z}^{(k-1)}$.

## 4 Empirical Evaluation

All implementations are made in TensorFlow and TensorFlow Probability (Abadi et al., 2015; Dillon et al., 2017).[1] On two synthetic datasets, TSC converges to near-optimal values. On survey data, TSC is more efficient than MSC and gives more reliable approximations on this task. For VAE, TSC achieves higher log-marginal likelihood on static MNIST, dynamic MNIST, and CIFAR10 than VAEs learned using four other baselines.

---

[1]Code is available at `link-temporarily-removed-for-anonymity`.

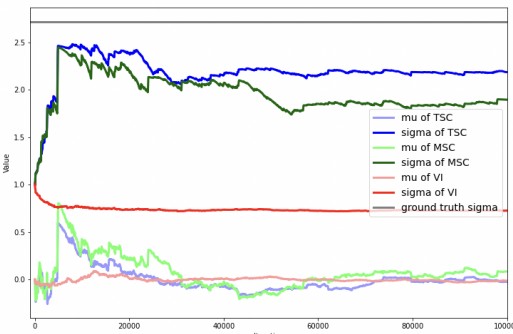

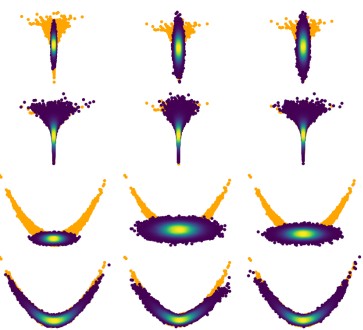

Figure 2: Variational parameters in TSC, MSC, and ELBO VI across iterations, using the Diagonal Gaussian family. The fitted Gaussian distributions approximate the funnel distribution. The plot shows parameter values of the first dimension (i.e. the horizontal dimension) of the distributions. Ground truth $\sigma$ is also drawn, but ground truth $\mu = 0$ is not drawn here. *TSC, while slightly more volatile than ELBO maximization, converges to parameter values closer to the ground truth.*

Figure 3: Synthetic targets (orange) with fitted posteriors laid on top. Top two: funnel; bottom two: banana. Rows 1 & 3: Gaussian family; rows 2 & 4: IAF family. Left: VI; middle: MSC; right: TSC. *In general, TSC more accurately approximates the posterior distribution.*

## 4.1 Synthetic Data

**Neal's Funnel Distribution.** We first study the funnel distribution described by Neal (2003), a distribution known to be hard to sample from by HMC. Let random variable $z$ have probability density function

$$p(z) = \mathcal{N}(z_1|0, 1)\mathcal{N}(z_2|0, e^{z_1/2}).$$

Then, $z$ follows the Funnel distribution.

**Banana Distribution.** Following Haario et al. (1999), we twist the Gaussian distribution to create a banana-shaped distribution. Let $(v_1, v_2) \sim \mathcal{N}(0, \left(\begin{smallmatrix} 100 & 0 \\ 0 & 1 \end{smallmatrix}\right))$, we transform $(v_1, v_2)$ with,

$$z_1 = v_1,$$
$$z_2 = v_2 + b \cdot v_1^2 - 100b,$$

where $b$ is a factor set to 0.02. Then, $(z_1, z_2)$ follows the Banana distribution.

Both distributions are visualized in Figure 3. We use the Adam optimizer (Kingma & Ba, 2015) with inverse time decay, decay rate $3 \cdot 10^{-4}$, and initial learning rate $3 \cdot 10^{-3}$. The HMC sampler consists of 1 chain, with step size $s$ tuned in $[0.03, 1)$ to target 67% acceptance rate, and number of leapfrog steps $L$ set to $\lceil \frac{1}{s} \rceil$. The HMC hyperparameter tuning follows the practice of Gelman et al. (2013).

**Results.** For the first variational family, we consider a diagonal Gaussian, $q(\theta) = N(\theta|\mu, \sigma^2 I)$. The optimal variational parameter for TSC is the true mean and standard deviation of $\theta$.

Figure 2 show variational parameters of the two dimensions by iteration. While both VI and TSC converge to near-optimal values for $\mu$, VI significantly underestimates uncertainty by converging to low values of $\sigma$. This problem is ameliorated by TSC, which gives $\sigma$ estimates much closer to the ground truth.

Table 1: Uncertainty estimation by the IAF family on synthetic data. The table gives standard deviation (std) across 100 groups, each group containing $10^7$ i.i.d. samples from fitted posteriors. In parentheses are standard errors across the 100 groups. *All methods give reasonable approximations using an expressive distribution, but TSC more closely recovers the true target distribution.*

| (a) Funnel distribution. | | | (b) Banana distribution. | | |
|---|---|---|---|---|---|
| Method | Std on Dim 1 | Std on Dim 2 | Method | Std on Dim 1 | Std on Dim 2 |
| Ground truth | 2.718 | 1 | Ground truth | 10 | 3 |
| ELBO VI | 2.286 (0.002) | 0.989 (0) | ELBO VI | 9.511 (0.002) | 2.675 (0.001) |
| MSC | 2.151 (0.001) | 0.961 (0) | MSC | 9.661 (0.002) | 2.562 (0.001) |
| **TSC** | **2.426 (0.002)** | **0.991 (0)** | **TSC** | **9.949 (0.002)** | **2.883 (0.001)** |

Figure 4: Estimates by states, where states are ordered by Republican vote in the 2016 election. We show TSC, 5000 sample unadjusted estimates, and 60000 sample unadjusted estimates. The unadjusted estimates are found by caluclating the mean and standard error of Bernoulli random variables. 95% confidence intervals are plotted. The closer to green the better. *TSC is robust to noise in the data sample and improves over the 5000 sample unadjusted estimate.*

As a second variational approximation, we use an IAF with 2 hidden layers. With an expressive posterior, each method gives reasonable approximations (Figure 3). Table 1 quantitatively compares these methods on synthetic data by giving standard deviations of large numbers of samples from the fitted IAF posteriors, and estimates from TSC are closest to the ground truth.

However, TSC still gives approximations that are often a little narrower than the true target distribution. One reason is the difficulty of the HMC chain to sample from certain areas in the target distribution. While an expressive flow further simplifies the geometry for HMC, it still does not guarantee perfect approximation in finite time.

## 4.2  Survey Data

We use the multilevel regression and post-stratification (MRP) model from Lopez-Martin et al. (2021) and apply it to a survey dataset provided by the same authors. Details of the model are given in Supplement A.2.1. The dataset originally comes from the 2018 Cooperative Congressional Election Study (Schaffner et al., 2019). The survey response is a binary variable representing individual answers to the question of

Table 2: The sum of squared difference from MRP MCMC estimates of mean and std of the response variable, one row for each method. *Lower is better.*

| Method | Mean Difference | Std Difference |
|---|---|---|
| ELBO VI | $1.18 \cdot 10^{-3}$ | $1.95 \cdot 10^{-3}$ |
| MSC | $2.44 \cdot 10^{-2}$ | $1.53 \cdot 10^{-2}$ |
| **TSC** | $\mathbf{5.86 \cdot 10^{-4}}$ | $\mathbf{1.02 \cdot 10^{-3}}$ |

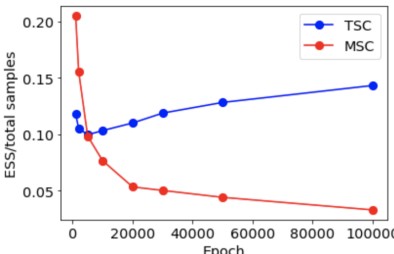

Figure 5: Cumulative ESS of TSC and MSC over number of samples across epochs. Higher is better. *As the transport map is learnt, TSC achieves higher ESS.*

whether to allow employers to decline coverage of abortions in insurance plans (Support / Oppose). Each response is attached with a set of features about the individual. The dataset consists of 60,000 data-points, but as suggested by the study, inference methods are trained on a randomly selected 5,000 subset. Reliable estimations are demonstrated by the ability to generalize from the 5,000 subset to the full 60,000 set, and closeness to gold-standard MRP MCMC results (Lopez-Martin et al., 2021).

We implement MSC and TSC with diagonal Gaussian approximations, and use the Adam optimizer with inverse time decay, decay rate $10^{-3}$, and initial learning rates 0.01. The HMC sampler has 1 chain, with step size $s$ tuned in $[0.03, 1]$ to target 67% acceptance rate and number of leapfrog steps $L$ set to $\lceil \frac{1}{s} \rceil$.

Figure 4 shows estimates by individuals' U.S. state, with states ordered by Republican vote share. The large-sample (60,000) estimates show an upward trend, which is intuitive. The estimates for TSC comes from 10,000 posterior samples. Figure 4 shows that TSC gives reasonable approximations since it generalizes from the 5,000 data points and gives estimates that are robust against noise in the data sample and are close to 60,000 sample estimates.

Asymptotic sample behavior measured through effective sample size (ESS) shows that the warped HMC chain underlying TSC outperforms the vanilla HMC chain used by MSC (Figure 5). It also suggests that dynamic training of the transport map actually hurts HMC efficiency when the variational approximation is still poor, but it quickly catches up when the approximation is better trained.

ELBO VI and MSC also provide reasonable approximations, but TSC is closer to MRP MCMC. We quantitatively compare TSC, ELBO VI, and the gold-standard MRP MCMC estimates (Lopez-Martin et al., 2021). Table 2 shows that TSC results are closer to MRP MCMC results than VI is to MRP MCMC.

### 4.3 Variational Autoencoders

Finally, we study TSC with amortized inference on statically binarized MNIST, dynamically binarized MNIST, and CIFAR10. With $D_{KL}(p||q)$ and dynamic updates of transport maps, a continuously run TSC is able to achieve higher log-marginal likelihood than several benchmarks.

**Implementation Details.** For benchmark methods, we use ELBO VI (Kingma & Welling, 2014; Rezende et al., 2014), importance-weighted (IW) autoencoder (Burda et al., 2016), MSC with the conditional importance sampler (CIS-MSC) (Naesseth et al., 2020), and NeutraHMC that follows the training procedure detailed in Hoffman (2017); Hoffman et al. (2019). We use the Adam optimizer with learning rates 0.001 and mini-batch size 256. Inference methods share the same architecture, which is detailed in Supplement A.2.2. For MNIST, we use a small-scale convolutional architecture and output Bernoulli parameters; for CIFAR10, we use a DCGAN-style architecture (Radford et al., 2016) and output Gaussian means.

Hoffman (2017) gives insightful training techniques: we also add an extra shearing matrix in the generative network and adapt HMC step-sizes $s$ to target a fixed acceptance rate. The best target acceptance rate is hand-tuned in [0.67, 0.95]. Number of leapfrog steps $L$ is set to $\lceil \frac{1}{s} \rceil$. The HMC initial state $\mathbf{z}^{(0)}$ is sampled from the encoder, whether it is previously warmed up or not. Additionally, TSC is more computationally demanding compared with ELBO maximization because of the HMC steps. We cap $L$ to 4 to ensure similar run-time with NeutraHMC, because TSC tends to lead to smaller step-sizes.

For TSC and NeutraHMC, we use one HMC step per data-point per epoch. For IWAE and CIS-MSC, we use 50 samples, suggested by Burda et al. (2016).

We estimate test log-marginal likelihood $\log p(\mathbf{x}; \theta)$ using Annealed Importance Sampling (AIS) (Neal, 2001; Wu et al., 2017) with 10 leapfrog steps and adaptive step sizes tuned to 67% acceptance, and 2500 annealing steps for MNIST, 7500 annealing steps for CIFAR10.

Table 3: Test log-marginal likelihood. 'Dim.' refers to latent dimensions; dim. 2 corresponds to Gaussian posterior, and dim. 64 or 128 corresponds to RealNVP posterior. -W: warm-up on the encoder previous to training. *: -2900 must be added to $\log p(\mathbf{x})$ for each CIFAR10 result. *TSC gives better predictive performance.*

| (a) Static MNIST | | | (b) Dynamic MNIST | | | (c) CIFAR10 | | |
|---|---|---|---|---|---|---|---|---|
| Dim. | Method | $\log p(\mathbf{x})$ | Dim. | Method | $\log p(\mathbf{x})^*$ | Dim. | Method | $\log p(\mathbf{x})^*$ |
| 2 | ELBO VI | $-133.94$ | 64 | ELBO VI | $-83.7$ | 128 | ELBO VI | $-34.61$ |
| | NeutraHMC-W | $-129.11$ | | IW | $-82.15$ | | IW | $-33.25$ |
| | **TSC** | $\mathbf{-128.88}$ | | NeutraHMC-W | $-83.01$ | | NeutraHMC-W | $-33.33$ |
| 64 | ELBO VI | $-61.01$ | | CIS-MSC | $-85.1$ | | CIS-MSC | $-33.38$ |
| | IW | $-58.84$ | | **TSC** | $\mathbf{-82.07}$ | | **TSC** | $\mathbf{-31.23}$ |
| | NeutraHMC-W | $-59.62$ | | | | | | |
| | CIS-MSC | $-60.4$ | | | | | | |
| | **TSC** | $\mathbf{-57.97}$ | | | | | | |

**Results based on log-marginal likelihood.** TSC achieves higher log-marginal likelihood with both low dimensional latent variables on Gaussian warped space and high dimensional latent variables on Real NVP warped space (Table 3). We use RealNVP instead of IAF for fast inversion $T_\lambda^{-1}(\mathbf{z})$. Two RealNVPs are stacked to form the variational posterior, each one having two hidden layers. Every model, including baselines, uses this flow distribution, contains a single layer of latent variables, and trains for 500 epochs.

Table 4: Three VAE metrics: downstream linear classification accuracy based on linear classifier (Acc), mutual information (MI), and number of active units (AU). The models come from corresponding ones in Table 3. Higher is better for each metric. *TSC gives best performance on CIFAR10, and NeutraHMC gives best performance on MNIST. CIS-MSC and TSC achieves the most number of active units.*

| | (a) Static MNIST | | | | | (b) CIFAR10 | | | |
|---|---|---|---|---|---|---|---|---|---|
| Dim. | Method | Acc | MI | AU | Dim. | Method | Acc | MI | AU |
| | ELBO VI | 96.25% | 11.58 | 48 | | ELBO VI | 44.5% | 9.2 | 62 |
| | IW | 96.7% | 10.25 | 54 | | IW | 45.06% | 9.07 | 59 |
| 64 | **NeutraHMC-W** | **97.16**% | **11.67** | **64** | 128 | NeutraHMC-W | 45.24% | 9.1 | 66 |
| | CIS-MSC | 94.48% | 10.04 | **64** | | CIS-MSC | 45.44% | 9.31 | **128** |
| | TSC | 95.76% | 10.61 | **64** | | **TSC** | **52.36**% | **10.73** | **128** |

TSC demonstrates effective synergy between transport map training and HMC sampling by training both encoders and decoders from scratch. This framework no longer requires a separate pretraining, which NeutraHMC does by warming up the encoder (which includes the normalizing flow) with ELBO maximization for 500 epochs. NeutraHMC then continues to train the warmed-up encoder during the main training phase (500 more epochs), as done in Hoffman (2017). Meanwhile, in the first 10 of the 500 TSC training epochs, the encoder is not trained, a design that improves stability.

**Additional evaluation metrics.**    The quality of the approximate posterior is reflected in its learned latent representations, and we use three additional metrics to evaluate these latent representations: downstream classification accuracy based on a linear classifier (Acc), mutual information (MI), and number of latent units (AU). The definition and implementational details of these metrics are given in Supplement A.2.2.

The metrics are computed for models trained on the static MNIST and CIFAR10 datasets (Table 4). The latent representations allow the linear classifiers to achieve high accuracy in general on MNIST, with NeutraHMC performing best on theeese three metrics on MNIST, even though its log marginal likelihood is lower than that of TSC. Meanwhile, TSC significantly outperforms all benchmarks in terms of the three additional metrics on CIFAR10. TSC and MSC, the methods that keep running the Markov chain and train parameters by completely relying on MCMC or HMC samples, have full number of active units on both datasets.

**Ablation Studies.**    Since we utilize both $D_{KL}(p||q)$ and a continuously-run, warped-space HMC, we wish to know whether the algorithm is as effective if one of these two components is removed. In the case of 2-dimensional latent variables, we first train a model with maximum likelihood using warped space HMC like in TSC, but it uses a pretrained encoder and no longer does $D_{KL}(p||q)$ to update the encoder. Next, we train a model that, compared to TSC, uses an ordinary HMC kernel without the space transformation. Results detailed in Supplement A.2.2 show that neither model achieves competitive performance. Therefore, not only is warped space HMC necessary for effective performance, but the dynamic $D_{KL}(p||q)$ updates of the approximate posterior and hence the transport map also play an essential role.

## 5    Conclusions

We develop Transport Score Climbing, improving VI with $D_{KL}(p||q)$ by using an HMC kernel on a simpler geometry defined by a transport map. This framework naturally leverages synergies since the transformation that warps the geometry is updated by HMC samples at each iteration, enabling more effective HMC sampling in future iterations. We illustrate the advantages of this method on two synthetic examples, survey data, and MNIST and CIFAR10 using VAE.

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

# A   Appendix

## A.1   Convergence of TSC

The TSC parameter estimates, $\lambda$ and $\theta$, may converge to a local optima of the forward KL and marginal likelihood, respectively. We formalize this result and detail the conditions in Proposition 1 for $\xi = (\theta, \lambda)$. The proposition is an *application* of (Gu & Kong, 1998, Theorem 1) which relies on (Benveniste et al., 1990, Theorem 3.17, page 304), and an adaptation of the result by Naesseth et al. (2020) that focuses only on $\lambda$.

*Proposition* 1. Assume A1-6, defined in the Supplement. If $\xi_k = (\theta_k, \lambda_k)$ for $k \geq 1$, defined in Algorithm 1, is a bounded sequence that almost surely visits a compact subset of the domain of attraction of $\xi^\star = (\theta^\star, \lambda^\star)$ infinitely often, then

$$\xi_k \to \xi^\star, \quad \text{almost surely.}$$

The proposition is an adaptation of Gu & Kong (1998, Theorem 1) based on Benveniste et al. (1990, Theorem 3.17, page 304) and a minor extension of Naesseth et al. (2020, Proposition 1). Let $\xi^\star = (\theta^\star, \lambda^\star)$, where $\theta^\star$ is a maximizer of the log-marginal likelihood and $\lambda^\star$ a minimizer of the forward KL divergence. Consider the ordinary differential equation (ODE), for $\xi(t) = (\theta(t), \lambda(t))$, defined by

$$\frac{\mathrm{d}}{\mathrm{d}t}\xi(t) = \begin{pmatrix} \mathbb{E}_{p(\mathbf{z}|\mathbf{x};\theta)}[\nabla_\theta \log p(\mathbf{z}, \mathbf{x}; \theta(t))] \\ \mathbb{E}_{p(\mathbf{z}|\mathbf{x};\theta)}[\nabla_\lambda \log q(\mathbf{z}; \lambda(t))] \end{pmatrix}, \; \xi(0) = \xi_0, \tag{8}$$

and its solution $\xi(t)$ for $t \geq 0$. If $\xi(t) = \widehat{\xi}$ is an unique solution to eq. 8 for $\xi_0 = \widehat{\xi}$, we call $\widehat{\xi}$ a stability point. The optima $\xi^\star$ is a stability point for eq. 8. We call the set $\Xi$ a domain of attraction of $\widehat{\xi}$, if the solution of eq. 8 for $\xi_0 \in \Xi$ remains in $\Xi$ and converges to $\widehat{\xi}$. Suppose that $\Xi$ is an open set in $\mathbb{R}^{d_\xi}$ and that $\xi_k \in \mathbb{R}^{d_\xi}$. Further, suppose $\mathbf{z_k} \in \mathbb{R}^{d_z}$ and that Z is an open set in $\mathbb{R}^{d_z}$. Denote the Hamiltonian Markov kernel used in TSC by $H_\xi(\mathbf{z}, \mathrm{d}\mathbf{z}')$, and repeated application of this kernel $H_\xi^k(\mathbf{z}, \mathrm{d}\mathbf{z}') = \int \cdots \int H_\xi(\mathbf{z}, \mathrm{d}\mathbf{z_1}) \cdots H_\xi(\mathbf{z_{k-1}}, \mathrm{d}\mathbf{z}')$. The length of the vector $\mathbf{z}$ is denoted by $|\mathbf{z}|$. Let $Q$ be any compact subset of $\Xi$, and $q > 1$ a sufficiently large (real) number so that the following assumptions holds. Like Gu & Kong (1998) we assume:

*A* 1.  The step size sequence satisfies $\sum_{k=1}^\infty \alpha_k = \infty$ and $\sum_{k=1}^\infty \alpha_k^2 < \infty$.

*A* 2 (Integrability).  There exists a constant $c_1$ such that for any $\xi \in \Xi$, $\mathbf{z} \in Z$ and $k \geq 1$

$$\int (|\mathbf{z}|^q + 1) H_\xi^k(\mathbf{z}, \mathrm{d}\mathbf{z}') \leq c_1 (|\mathbf{z}|^q + 1)$$

*A* 3 (Markov Chain Convergence).  Let $p(\mathbf{z}|\mathbf{x};\theta)$ be the unique invariant distribution for $H_\xi$. For each $\xi \in \Xi$

$$\lim_{k\to\infty} \sup_{\mathbf{z}\in Z} \frac{\int (|\mathbf{z}'|^q + 1) |H_\xi^k(\mathbf{z}, \mathrm{d}\mathbf{z}') - p(\mathrm{d}\mathbf{z}'|\mathbf{x};\theta)|}{|\mathbf{z}|^q + 1} = 0$$

*A* 4 (Continuity in $\xi$).  There exists a constant $c_2$ such that for all $\xi, \xi' \in Q$,

$$\int (|\mathbf{z}'|^q + 1) |H_\xi(\mathbf{z}, \mathrm{d}\mathbf{z}') - H_{\xi'}(\mathbf{z}, \mathrm{d}\mathbf{z}')| \leq c_2 |\xi - \xi'| (|\mathbf{z}|^q + 1)$$

*A* 5 (Continuity in **z**). There exists a constant $c_3$ such that for all $\mathbf{z}_1, \mathbf{z}_2 \in Z$

$$\sup_{\xi \in \Xi} \left| \int \left( |\mathbf{z}'|^q + 1 \right) \left( H_\xi(\mathbf{z}_1, d\mathbf{z}') - H_{\xi'}(\mathbf{z}_2, d\mathbf{z}') \right) \right| \le c_3 |\mathbf{z}_1 - \mathbf{z}_2| \left( |\mathbf{z}_1|^q + |\mathbf{z}_2|^q + 1 \right)$$

*A* 6 (Conditions on Gradients). For any compact subset $Q \subset \Xi$, there exists (positive) constants $p$, $k_1$, $k_2$, $k_3$, $\nu > \frac{1}{2}$ such that for all $\xi, \xi' \in \Xi$ and $\mathbf{z}, \mathbf{z}_1, \mathbf{z}_2 \in Z$

$$|\nabla_\xi \left( \log p(\mathbf{z}, \mathbf{x}; \theta) + \log q(\mathbf{z}; \lambda) \right)| \le k_1 \left( |\mathbf{z}|^{p+1} + 1 \right)$$

$$|\nabla_\xi \left( \log p(\mathbf{z}_1, \mathbf{x}; \theta) + \log q(\mathbf{z}_1; \lambda) \right) - \nabla_\xi \left( \log p(\mathbf{z}_2, \mathbf{x}; \theta) + \log q(\mathbf{z}_2; \lambda) \right)| \le k_2 |\mathbf{z}_1 - \mathbf{z}_2| \left( |\mathbf{z}_1|^p + |\mathbf{z}_2|^p + 1 \right)$$

$$|\nabla_\xi \left( \log p(\mathbf{z}, \mathbf{x}; \theta) + \log q(\mathbf{z}; \lambda) \right) - \nabla_\xi \left( \log p(\mathbf{z}, \mathbf{x}; \theta') + \log q(\mathbf{z}; \lambda') \right)| \le k_3 |\xi - \xi'|^\nu \left( |\mathbf{z}|^{p+1} + 1 \right)$$

The results follows from Gu & Kong (1998, Theorem 1), under assumptions A1-6, by identifying:

$$\theta = \xi$$
$$x = \mathbf{z}$$
$$\Pi_\theta = H_\xi$$
$$H(\theta, x) = \nabla_\xi \left( \log p(\mathbf{z}, \mathbf{x}; \theta) + \log q(\mathbf{z}; \lambda) \right)$$

and $\Gamma_k = I$, $I(\theta, x) = 0$ where left is their notation and right is our notation.

## A.2 Experiments

### A.2.1 Survey Data

**Model.** Following (Lopez-Martin et al., 2021), we model binary variable **x** taking values 0 or 1 with a multilevel regression model. **x** indicate individual responses, and each individual comes with given features: state, age, ethnicity, education, and gender. For each data-point $x_i$, the model is defined as,

$$p(x_i = 1) = \text{logit}^{-1}(z_{s[i]}^{\text{state}} + z_{a[i]}^{\text{age}} + z_{r[i]}^{\text{eth}} + z_{e[i]}^{\text{edu}} + z_{g[i],r[i]}^{\text{gen.eth}} + z_{e[i],a[i]}^{\text{edu.age}} + z_{e[i],r[i]}^{\text{edu.eth}} + \beta^{\text{gen}} \cdot \text{Gen}_i).$$

$$z_s^{\text{state}} \sim \mathcal{N}(\gamma^0 + \gamma^{\text{south}} \cdot \mathbb{1}(\text{s in south}) + \gamma^{\text{northcentral}} \cdot \mathbb{1}(\text{s in northcentral}) + \gamma^{\text{west}} \cdot \mathbb{1}(\text{s in west}), \sigma^{\text{state}}), \text{for } s = 1, ..., 50,$$

$$z_a^{\text{age}} \sim \mathcal{N}(0, \sigma^{\text{age}}), \text{for } a = 1, ..., 6,$$

$$z_r^{\text{eth}} \sim \mathcal{N}(0, \sigma^{\text{eth}}), \text{for } r = 1, ..., 4,$$

$$z_e^{\text{edu}} \sim \mathcal{N}(0, \sigma^{\text{edu}}), \text{for } e = 1, ..., 5,$$

$$z_{g,r}^{\text{gen.eth}} \sim \mathcal{N}(0, \sigma^{\text{gen.eth}}), \text{for } g = 1, 2 \text{ and } r = 1, ..., 4,$$

$$z_{e,a}^{\text{edu.age}} \sim \mathcal{N}(0, \sigma^{\text{edu.age}}), \text{for } e = 1, ..., 5 \text{ and } a = 1, ..., 6,$$

$$z_{e,r}^{\text{edu.eth}} \sim \mathcal{N}(0, \sigma^{\text{edu.eth}}), \text{for } e = 1, ..., 5 \text{ and } r = 1, ..., 4.$$

Each $z_*^*$ is a latent variable. For example, $z_:^{\text{state}}$ is a length-50 latent variable that indicates the effect of state on the binary response. As another example, $z_{:,:}^{\text{edu.age}}$ indicates the interaction effect of education and age, and is length-30 because there are 5 education levels and 6 age levels. In total, the model has a length-123 latent variable **z**. We model the rest, namely $\gamma^*$, $\sigma^*$, and $\beta^{\text{gen}}$, as model parameters where we find fixed estimates.

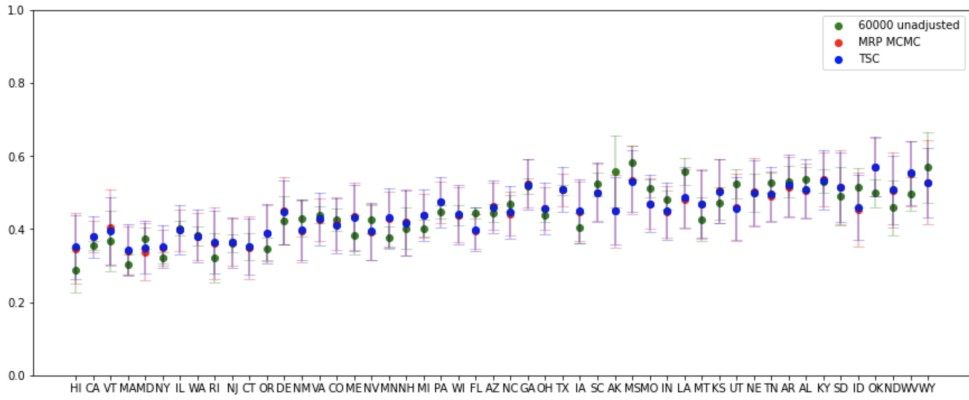

(a) TSC, MRP MCMC, and 60000 sample unadjusted estimate.

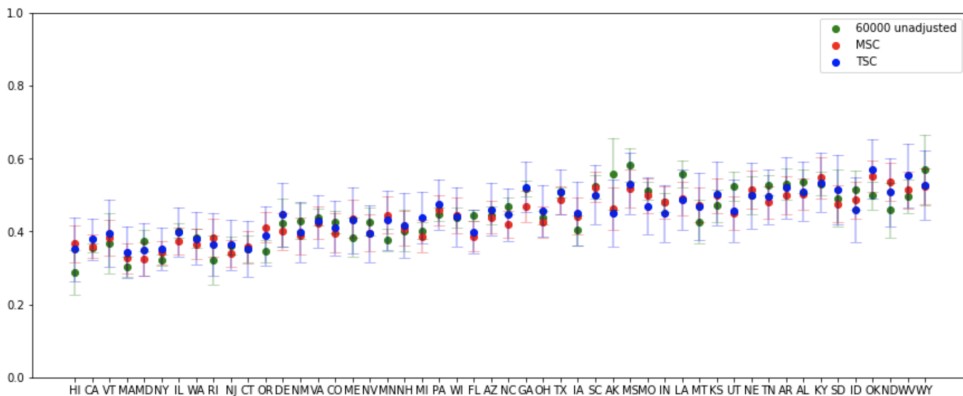

(b) TSC, MSC, and 60000 sample unadjusted estimate.

Figure 6: Estimates by states, where states are ordered by Republican vote in the 2016 election. 95% confidence intervals are plotted. The closer to green the better.

**Results.** Reliable approximations on the survey data are shown by ability to generalize from small, 5,000 sample to the full 60,000 sample, and closeness to gold-standard MRP MCMC results. We visualize TSC, MSC, and MRP MCMC estimates by state. Figure 6a shows that the mean of TSC estimates are barely discernible from the mean of results given by the gold-standard MRP MCMC (Lopez-Martin et al., 2021). Figure 6b shows that MSC is also robust against noise from the small 5,000 sample, but it slightly differs in results from TSC.

### A.2.2   Variational Autoencoder

**Architecture.**   In MNIST, both statically and dynamically binarized, the encoder uses two convolutional layers with number of filters 32 and 64, followed by a dense layer that outputs Gaussian mean and log-variances (so its hidden-size is two times latent variable dimension). The decoder begins with a dense layer with hidden-size $7 \cdot 7 \cdot 32$, followed by three transpose convolutional layers with number of filters 32, 64, and

1, and it outputs a Bernoulli parameter for each pixel. All layers use kernel size 3, stride size 2, same padding, and ReLU activations, except for the last transpose convolutional layer that uses stride size 1.

A DCGAN-style architecture is used for CIFAR10, featuring no dense layers, batch normalization, and leaky ReLU. The encoder uses four convolutional layers with number of filters 64, 128, 256, and latent dimension times 2. The last layer has no activation function and is flattened to give Gaussian mean and log-variances. The decoder uses four transpose convolutional layers with number of filters 256, 128, 64, 3. The last layer uses tanh activation and outputs Gaussian mean. Batch normalization and leaky ReLU (0.2) are applied after each layer except for the last layer in encoder and decoder. All layers use kernel size are 4, stride size 2, and same padding, except that the last layer in encoder and first layer in decoder use stride size 1 and valid padding.

**Additional evaluation metrics.** In addition to log-marginal likelihood, we use three additional metrics to evaluate the quality of the approximate posterior and its latent representations. The metrics are defined as follows.

*Downstream classification accuracy.* The approximate posterior maps every data-point to a vector representation. We take these vectors that correspond to training data-points from each method, and for each method we train a linear classifier based on the vector representations to predict the class of the data-point. Accuracy is reported on held-out test dataset.

*Mutual information (MI).* MI $I(\mathbf{z}; \mathbf{x})$ measures the dependence between data $\mathbf{x}$ and latent representation $\mathbf{z}$. High MI is desirable because it suggests that the generative model makes use of unique information encoded by latents $\mathbf{z}$. MI is defined as follows (Hoffman & Johnson, 2016; Dieng et al., 2019),

$$I(\mathbf{z}; \mathbf{x}) = \mathbb{E}_{\widehat{p}(\mathbf{x})}\big[D_{KL}(q(\mathbf{z}|\mathbf{x}; \lambda)||p(\mathbf{z})) - D_{KL}(q(\mathbf{z}; \lambda)||p(\mathbf{z}))\big],$$

where $q(\mathbf{z}; \lambda)$ is the 'aggregate posterior', a marginal over the latent $\mathbf{z}$ defined as $q(\mathbf{z}; \lambda) = \frac{1}{N}\sum_{i=1}^{N} q(\mathbf{z}|\mathbf{x}_i; \lambda)$, and $N$ is number of data-points. $I(\mathbf{z}; \mathbf{x})$ is approximated by Monte Carlo.

*Number of latent units (AU).* AU measures how many units on the latent representation are 'active' (Burda et al., 2016) (for instance, if a 64-dimensional latent is used, we measure how many units among the 64 total units are active). It is desirable that all latent variables are used. A latent dimension is considered active if $\text{Cov}_{\mathbf{x}}(\mathbb{E}_{u \sim q(u|\mathbf{x}; \lambda)[u]}) > 0.02$.

**Ablation studies.** We do two VAE ablation studies under the case of two dimensional latent variables.

Study I: no approximate inference with $\text{KL}(p||q)$; only maximum likelihood on $p(\mathbf{x}; \theta)$. First, we wonder whether warped space HMC itself along with a pretrained transport map achieves competitive performance. That is, the encoder is no longer trained, and the overall training is essentially Monte Carlo EM (Duane et al., 1987; Kingma & Welling, 2014). It achieves $-156.1$ log-marginal likelihood after the same number of epochs of training, lower than all baselines.

Study II: run HMC on original space instead of warped space. We also test whether running HMC on the original space together with approximate posterior training via $\text{KL}(p||q)$ achieves competitive performance. The estimated log-marginal likelihood is $-133.9$, lower than both TSC and NeutraHMC.

