# OpenReview forum: "Transport Score Climbing: Variational Inference Using Forward KL and Adaptive Neural Transport"
_TMLR — Rejected by TMLR_

### Review · Reviewer_H2tY · 2022-11-20

**Summary Of Contributions:**

This works study the problem of forward variation inference.
In the traditional variational inference, mainly we focus on revers KL divergence to find a surrogate distribution q through minimizing KL(q||p). In this work, the author study the minimization of forward KL, i.e., KL(p||q).

The forward KL is more natural to the commonly used reverse KL, however, it comes with intractable steps. It requires sampling from a distribution that we do not have access to.

To tackle this intractable step, prior work of Naesseth et al proposed using MCMC based methods to handle the computation.

The current work suggests using Hamiltonian MC on a transport latent space following the developments in  Hoffman et al. (2019).

In the end, the authors show that their proposed method results in high likelihood when compared to other methods.



**Audience:**

Yes

**Broader Impact Concerns:**

This work is very useful when computing posterior accurately is very important.

**Claims And Evidence:**

Yes

**Requested Changes:**

Please make sure the writing is highly improved.
Please also show how this model does not just in terms of log(x) but other metrics. Also comparison with many non VI methods are missing.

**Strengths And Weaknesses:**

Strength.
The proposed method provides improvements in likelihood when we are concerned with maximum likelihood.

Weakness:
- The presentation of the paper can be significantly improved.
- The grammar can also be improved.
- Many things are not defined. What is M in the Hamiltonian, and what is p(x|z)? What is z?
- There is almost no attempt to motivate this paper and this study. Why do we care about the posterior of z, especially when it is what we make ourselves?
- The notation can be improved and explained. Use of epsilon and z_0. The updates in the HMC sections are written as equations. What are those tilde variables? What is a state in HMC? these are not defined. Please make the paper rigorous.
- Another example, x is used for variables and datasets. Please be more concerned about the setup. There are many more such notation loosenesses.
- It is not clear how the HMC updates are computed, we don't have p(x,z).
- How the logp(x) is computed? for what x?

---

> ### Author Response · Authors · 2022-12-14
> **Response to reviewer H2tY**
>
> We want to thank the reviewer for devoting their time and expertise. The revised paper has been uploaded.
>
> Q: The presentation of the paper can be significantly improved. The grammar can also be improved.\
> A: We will keep this in mind as we revise the paper. Are there any concrete examples we should pay particular attention to?
>
> Q: Many things are not defined. What is M in the Hamiltonian, and what is p(x|z)? What is z?\
> A: The mass matrix M of the Hamiltonian dynamics is the identity matrix (since we assume momentum matrix comes from standard normal). We updated section 2.2 to clarify.
>
> z is defined in the first sentence of section 2 ‘Background’, and it is the latent (unknown/unobserved) variables of the probabilistic model p(x,z). p(x|z) is the likelihood of the probabilistic model p(x,z) and is mentioned in the first paragraph in section 3.3. For clarity we now define p(x,z), p(x|z), p(z) all under section 2 ‘Background’.
>
> Q: There is almost no attempt to motivate this paper and this study. Why do we care about the posterior of z, especially when it is what we make ourselves?\
> A: Posterior inference is one of the main goals of probabilistic modeling as mentioned in the first paragraph of the Section 1, Introduction. It allows us to compute the data marginal likelihood, which is relevant in many types of models (Bayesian regression, Bayesian neural network, VAE, etc.), make predictions about latent (unknown) variables based on data, etc. Furthermore, if the model is interpretable, the posterior lets us understand latent factors behind the data.
>
> Q: The notation can be improved and explained. Use of epsilon and z_0. The updates in the HMC sections are written as equations. What are those tilde variables? What is a state in HMC? these are not defined. Please make the paper rigorous.\
> A: Epsilon and z_0 are indeed different. Epsilon always comes from a standard normal distribution (which is also the flow’s base distribution; section 3.1), z_0 distribution is defined by: z ~ p(z|x) (true posterior), and z_0 = inverse-flow(z) (section 3.2).
>
> The equations in background HMC section 2.2 define one “leapfrog” step. The z and m set in the last leapfrog step were denoted by z_tilde and m_tilde. A state in HMC is the latent variable z generated at a given iteration, along with momentum variable m. We updated section 2.2 to clarify, having removed the use of tildes and written mathematical equations instead of algorithmic equations.
>
> Q: Another example, x is used for variables and datasets. Please be more concerned about the setup. There are many more such notation loosenesses.\
> A: In this paper, we use z for latent variables and x for data, which are simultaneously random variables (section 2 Background).
>
> Q: It is not clear how the HMC updates are computed, we don't have p(x,z).\
> A: We define the target of HMC in the methods section (equation (3) of section 3.2). p(x,z) is known because it is the product of the likelihood p(x|z) and the prior p(z). We have updated the section to clarify.
>
> Q: How is the logp(x) computed? for what x?\
> A: As described in Section 4.3, paragraph 5, we follow the approach introduced by Wu et al. (2017), also used by Hoffman et al. (2019) (NeutraHMC), and estimate the test log-marginal likelihood using annealed importance sampling.
>
> Q: Please also show how this model does not just in terms of log(x) but other metrics.\
> A: Estimating the log-marginal likelihood log p(x) for a held-out test dataset is the standard way of evaluating VAE’s. While the NeutraHMC paper only shows marginal likelihood for VAE, we computed three new metrics recently: downstream classification accuracy with linear classifier, mutual information (MI), and number of active units (AU), detailed in section 4.3 in the revision. Here is a summary: the latent representations allow the linear classifiers to achieve high accuracy in general on MNIST, with NeutraHMC with warm up performing best on these three metrics on MNIST, even though its log marginal likelihood is lower than that of TSC. Meanwhile, TSC significantly outperforms all benchmarks in terms of these three metrics on CIFAR10. TSC and MSC, the methods that keep running the Markov chain and train parameters by relying on MCMC or HMC samples, have the highest number of active units on both datasets.
>
> References\
> Yuhuai Wu, Yuri Burda, Ruslan Salakhutdinov, and Roger B. Grosse. On the quantitative analysis of decoder-based generative models. ArXiv, abs/1611.04273, 2017.\
> Matthew D. Hoffman, Pavel Sountsov, Joshua V. Dillon, Ian Langmore, Dustin Tran, and Srinivas Vasudevan. Neutra-lizing
> bad geometry in Hamiltonian Monte Carlo using neural transport. arXiv:1903.03704, 2019.

---

### Review · Reviewer_kw6Z · 2022-11-27

**Summary Of Contributions:**

The authors propose a combination of (warped) HMC and variational inference designed to minimize the "forward" KL divergence, that is $KL(p, q)$ as well as to perform maximum likelihood. This is an adaption of the Markovian Score Climbing algorithm proposed in Naesseth et al 2020, which takes a similar approach with a different MCMC scheme. The map used to warp the space is adaptive, with the hope of achieving faster mixing in HMC. The authors demonstrate the method on several toy problems, a Bayesian survey data problem and a variational auto-encoder.

**Audience:**

Yes

**Broader Impact Concerns:**

The paper focuses on improved methods for probabilistic inference. As the paper is primarily methodological, and the method does not obviously lend itself to ethically ambiguous applications (any more than other methods of probabilistic inference). The datasets used for the variational autoencoder (MNIST, CIFAR10) are both commonly used, and I am not aware of significant concerns with the use of either dataset. The survey data does not appear to raise significant ethical concerns.


**Claims And Evidence:**

No

**Requested Changes:**

## Statement of conditions in main text
- In the main text (section 3), some intuition for the types of conditions needed for TSC to converge to a local optimum should be given. As far as I can tell this is one of the primary contributions of the paper. I don't think it is necessary to state all of the conditions explicitly in the main text if you view them largely as technical, regularity conditions and see the primary contribution as empirical/methodological. However, giving a brief summary with intuition would still improve the paper.

## Convergence
- Typically, results in MCMC theory rely on using a fixed kernel with the "right" stationary distribution and establishing ergodicity to show convergence. In section 3.2, you state that your kernel changes between each iteration. It isn't at all clear to me that this leads to a convergent algorithm. Please comment on consistency of the method in terms of sampling from the target, and the resulting quality of estimation of the needed expecations.
- The claim that "the corresponding algorithm can be shown to maximize the true marginal likelihood exactly" (section 3.2.1) is much stronger than that it converges to a local optimum, which I think is what the authors argue in other parts of the text, as one would generally not expect the marginal liklihood to have a unique local optimum.

## Experimental claims

### Synthetic experiments
- Under figure 3 the claims is made that "TSC more accurately approximates the posterior distribution". I don't find figure 3 to be a very compelling illustration of this, and don't think it really supports the claim. It does seem that it produces larger estimates of the posterior variance. This may or may not be a good thing depending on application, but I suspect there are applications where this is useful. The authors should either provide more convincing evidence that the posterior is "better" or narrow the claim to avoiding underestimating the posterior variance by as much, and give an example of an application where this is desirable.
- I don't understand what the "groups" in stable 1 are. Please clarify what the table is summarizing with some discussion in the main text.
- The synthetic experiments mostly compare the method to (reverse KL) VI on the basis of the first two moments when fitting Gaussian distributions. As observed earlier, if forward KL VI is done exactly, these should match the moments of the true posterior. Is there an application the authors have in mind where this is the "right" way to judge the success of the approximation? Otherwise this seems a bit of an unfair comparison since other properties of the posterior might be better captured by minimizing the reverse KL.
### Survey experiment
- I am unclear why I would use TSC as opposed to using the MRP MCMC method. Your metrics compare other approximations to this method. But why not just use MRP MCMC?
- Multilevel regression and post-stratification should be written out when first referring to MRP MCMC. Also a brief description of this algorithm would be useful.
- Many parameters must be selected for the method (learning rate, number of leapfrog steps, number of chains, step size, decay schedule...). While these are clearly stated, it should also be stated how they were selected. Where these the first parameters you tried? Or was some trial and error used? It would be good to know how robust the method is to the selection of these, as well as how well the authors expect "default" parameters to apply to new datasets the method is applied to.
## VAE experiment
- It isn't clear if the differences in the estimated marginal likelihood matter for any application. Something more concrete for showing performance gains for would be useful.

## Proof Correctness and revision
- In its current form, the proof is difficult to follow and I was not able to verify whether or not it is correct. The following changes should be made:
  - Style changes:
    - Move assumptions prior to the statement of the Theorem. Provide some description of whether or not they are reasonable for the problems considered. Ideally, verify the assumptions in a very simple case (e.g. with both distributions Gaussian).
    - Include a statement and ideally a proof sketch of the result quoted from Gu \& Kong for self-containedness.
  - Questions:
    - What is $\epsilon'$ in assumption 5?
    - Should $H$ be indexed by $q$ as well? I am concerned that the transition kernel changing during training has not been taken into account in the proof. Could the authors clarify the dependence of $h$ on $q$, and whether this leads to issues with assumption A3 or elsewhere in the proof?

## Minor

- The claim "if $\mathcal{Q}$ is a subset of the exponential family distributions then the moments of the optimal $q$ matches the moments of the posterior $p$ exactly." is not correct as stated. First, it implies all of the moments of $q$ will match those of $p$. Second, it depends on $\mathcal{Q}$ containing an element that can match certain moments of $p$, which is not true for arbitrary subsets of an exponential family. Please adjust the phrasing to narrow the claim so that it is correct.
- Font on Figure 4 should be larger. It is hard to read.
- Write out "with respect to". There is no need to save space e.g. above equation 2.
- Capitalization in bibliography should be fixed. For example,
  - monte carlo -> Monte Carlo
  - taylan. cemgil -> Taylan Cemgil
  - kullback-leibler -> Kullback-Leibler

**Strengths And Weaknesses:**

## Strengths
- The scope of experiments is good. In particular, I appreciate that the authors first illustrate the method on a few small examples where "ground truth" is known, before applying the method to two more realistic problems.

## Weaknesses
- Several statements are made in the main text that are not quite correct. These can be revised reasonably easily.
- The proof of the result in A.1 is not detailed enough to follow. The authors need to improve this prior to acceptance.

More complete feedback related to these weaknesses are discussed below.

---

> ### Author Response · Authors · 2022-12-14
> **Response to reviewer kw6Z**
>
> We want to thank the reviewer for devoting their time and expertise. The revised paper has been uploaded.
>
> Q: The proof of the result in A.1 is not detailed enough to follow. \
> A: The result in A.1 is an application of the result by Gu & Kong (1998) and we claim no added novelty other than applying it to the TSC setting. For the full proof details we refer to that paper.
>
> Q: Intuition for the types of conditions needed for TSC to converge? \
> A: The primary contributions of the paper are empirical/methodological as stated in the introduction. A1 is a standard step-size requirement for stochastic approximation algorithms, A2-5 are technical conditions on the Markov kernel H(z,dz’) that are difficult to check in practice, and A6 are regularity conditions on the gradients of the user-chosen log p(x,z) and log q(z).
>
> Q: Typically, results in MCMC theory rely on using a fixed kernel. You state that your kernel changes between each iteration. \
> A: Indeed the TSC kernel changes at each iteration. The result by Gu & Kong (1998) specifically applies to this changing setting.
>
> Q: Exact maximization vs. local optimum: authors are actually arguing for local optimum. \
> A: The intention was to contrast the TSC approach, which tries to maximize the log-marginal likelihood directly, with reverse KL-based methods that maximize a surrogate lower bound to the log-marginal likelihood. We updated that line in section 3.2.1 to clarify.
>
> Experiments - synthetic. \
> Q: Larger estimates of the posterior variance may or may not be a good thing. \
> A: The main goal with the synthetic experiments is to illustrate the posterior uncertainty underestimation by reverse KL VI and contrast this to forward KL VI which does not suffer from this issue to the same degree. This underestimation leads to overconfidence in the posterior predictive distribution. Furthermore, it illustrates the convergence of TSC on synthetic datasets where ground truth is available.
>
> Q: What are the "groups" in table 1? \
> A: In each group, we take samples from the posterior distribution and compute their statistics. We repeat this 100 times to also compute a confidence interval for these statistics.
>
> Experiments - survey. \
> Q: With MRP MCMC as what other algorithms compare with - why TSC and why not just use MRP MCMC?
> A: A benefit of TSC over MRP MCMC is having a “distilled” approximate posterior distribution (normalizing flow), rather than having to save many MCMC samples. Having an approximate posterior has advantages such as allowing one to get large numbers of more samples very quickly.
>
> Q: Multilevel regression and post-stratification should be written out when first referring to MRP MCMC. Also a brief description would be useful. \
> A: We will update and write out the full name multilevel regression and post-stratification. MRP is the name for the model, and the model is detailed in supplement A.2.1. We have made this connection and definition clear in section 4.2.
>
> Q: While these are clearly stated, it should also be stated how they were selected. \
> A: For HMC, as suggested by Gelman, et al. (2013), we target a 67% acceptance rate, and find the HMC step-size that results in this rate, and leapfrog steps are always 1/step-size. For the sake of comparison all methods use only 1 chain. These are independent of the optimizer’s hyperparameters. For optimizer hyperparameters (learning rate, decay rate), we tune with grid search as usual. This applies to synthetic data as well. For VAE, HMC step-sizes are adaptive instead, and this choice follows previous practices and is detailed in section 4.3. We will improve the text in section 4 to make the description clearer.
>
> Experiments - VAE. \
> Q: It isn't clear if the differences in the estimated marginal likelihood matter for any application. Something more concrete for showing performance gains would be useful. \
> A: Estimating the log-marginal likelihood log p(x) for a held-out test dataset is the standard way of evaluating VAE’s. While the NeutraHMC paper only shows marginal likelihood for VAE, we computed three new metrics recently: downstream classification accuracy with linear classifier, mutual information (MI), and number of active units (AU), detailed in section 4.3 in the revision. Here is a summary: the latent representations allow the linear classifiers to achieve high accuracy in general on MNIST, with NeutraHMC with warm up performing best on these three metrics on MNIST, even though its log marginal likelihood is lower than that of TSC. Meanwhile, TSC significantly outperforms all benchmarks in terms of these three metrics on CIFAR10. TSC and MSC, the methods that keep running the Markov chain and train parameters by relying on MCMC or HMC samples, have the highest number of active units on both datasets.
>
> References\
> Gelman, A., Carlin, J.B., Stern, H.S., Dunson, D.B., Vehtari, A., & Rubin, D.B. (2013). Bayesian Data Analysis (3rd ed.). Chapman and Hall/CRC.

---

### Review · Reviewer_HvpZ · 2022-12-01

**Summary Of Contributions:**

The authors consider the task of optimizing the forward KL(p,q), which avoids the mode-seeking behaviour of KL(q,p), but comes with the price of requiring the posterior instead of being able to rely on an evidence lower bound approach. The authors follow prior work by Naesseth et al. (2020) who use an MCMC approach for this task. Their adaptation consists of switching to an HMC sampling method in a space warped by a normalizing flow to make the HMC more efficient. In that, they follow prior work by Hoffman et al. (2017,2019) who introduced a similar approach for the KL(q,p) setting.


**Audience:**

Yes

**Claims And Evidence:**

Yes

**Requested Changes:**

- Include NeutraHMC comparisons in the first two experiments (Sec 4.1/4.2), not just the third one. Similarly, for the visualizations in Fig 3 there should exist comparable plots for MSC and NeutraHMC (can be in the appendix for space reasons).
- Introduce stds to Table 2/3
- Fix the Reference section which is full of inconsistencies. E.g., on the first page, Andrieu is once referred to as _C. Andrieu_ and once as _Christope Andrieu_,... .


**Strengths And Weaknesses:**

## Strengths
The authors combine prior work in a novel way. They show that this combination (together with the minor extensions of learning the parameters of the normalizing flow at the same time) gives an improvement over these prior methods in a series of experiments.

## Weaknesses
The novelty is relatively minor and detailed comparisons with MSC and NeutraHMC are partially missing.

### Minor
- Q: What do the authors mean by statically vs dynamically binarized MNIST?
- Table 2/3 lack error bars
- The visualization of Fig 4 (and related in the appendix) makes it difficult to have a detailed view of the relative performance. Adding plots that visualize this instead of the absolute values would make visual comparison easier.
- Typo in the legend of Figure 2 HSC vs TSC

---

> ### Author Response · Authors · 2022-12-14
> **Response to reviewer HvpZ**
>
> We want to thank the reviewer for devoting their time and expertise. The revised paper has been uploaded.
>
> Q: What do the authors mean by statically vs dynamically binarized MNIST? \
> A: Description of the data from tensorflow datasets: dynamically binarized MNIST is “a specific binarization of the MNIST images originally used in (Salakhutdinov & Murray, 2008). This dataset is frequently used to evaluate generative models of images.”
>
> Q: Error bars for Table 2/3 \
> A:  The numbers provided in the tables are representative. We are currently re-running the experiments to generate error bars. However, due to time-constraints these will not be finished by 15 December. If accepted, we will update with the error bars.
>
> Q: The visualization of Fig 4 (and related in the appendix) makes it difficult to have a detailed view of the relative performance. Adding plots that visualize this instead of the absolute values would make visual comparison easier. \
> A: The survey dataset is chosen for its interpretability and its nature as real world data. This visualization aims at showing that TSC generates interpretable results on this model and dataset. We can make the dots appear on slightly different x-axis positions, so the three values and their confidence intervals in each graph can be more easily compared.
>
> Q: Include NeutraHMC comparisons in the first two experiments (Sec 4.1/4.2), not just the third one. Similarly, for the visualizations in Fig 3 there should exist comparable plots for MSC and NeutraHMC (can be in the appendix for space reasons). \
> A: The variational approximation to the posterior from NeutraHMC (whose main contribution is HMC not VI) is identical to one learnt by reverse KL, corresponding to VI in Fig. 2 and 3. The 4.1 and 4.2 experiments focus on comparing the learned approximations to the posterior distributions of VI methods (including MSC). We have added corresponding figures for MSC in the revision, and these are comparable with TSC.
>
> Q: Fix reference section inconsistencies. \
> A: We have fixed reference section inconsistencies in the updated version.

---

### Comment · Reviewer_kw6Z · 2022-12-15
**Revised Submission**

Could the authors provide a summary of changes made to the text in the newly uploaded version? I don't see the changes explicitly marked in the new pdf, and having at least a relatively complete summary of all of the changes made to the text would be helpful to me (and I suspect other reviewers) when reading back through the paper.

Also, as a formatting note, it looks to me as though something has gone wrong with the upload in terms of top and side margins (the side margins are under an inch, and the header leaves almost no top margin). Perhaps something went wrong with saving the pdf that caused this. It would be good if the authors could fix this.

---

> ### Author Response · Authors · 2022-12-16
> **Response to 'Revised Submission'**
>
> We thank the reviewer for noticing this error. The revision is updated and the current version fixes the formatting error.
>
> Here is a summary of changes from the first version:
> * The major change is the addition of three VAE evaluation metrics and their results (section 4.3).
> * Changes to and clarifications on presentation and notation based on review suggestions (section 2, section 2.2, section 4.2).
> * Reverting the claim of exact maximization to direct maximization and local optima convergence (section 3.2.1).
> * Adding MSC to the figure that originally has ELBO VI and TSC (section 4.1).
> * Fixing several inconsistencies in the reference section.

---

### Decision · Action_Editors · 2023-01-06

**Recommendation:** Reject

**Comment:**

We encourage the authors to resubmit and in  the resubmission please make proof can be made self-contained and provide an explanation of the conditions under which the claim holds. The main reason for not accepting the manuscript is the readability issues of the proof and the reviews did not consider the reply in the rebuttal and the updated version fixing it. The reviewers considered the contribution as small but worthy modification of the proof by Naesseth et al. (2020) and has directly inherited its readability problems. The reviewers think they in turn had inherit the conditions and their readability problems directly from the work by Gu and Kong (1998). They finally give some small comments on their motivation and their satisfiability in practice and traced back original proof given in the textbook by Benveniste et al. (1990). As a result the reviewers see the result worthy of publication by the standards of TMLR. We strongly encourage to resubmit and either make the submission purely experimental/methodological or make the proof checkable and self contained. Please also add clear discussion in the main text. Following earlier reviews, the paper will benefit from having a more in-depth discussion on motivations.

**Audience:**

The reviewer agreed that if the proof can be made self-contained so it can be checked, that there is a contribution that give a minor but consistent improvement over the work of Naesseth et al.

**Claims And Evidence:**

The reviewers have noted that the paper does not have a proof that is sufficiently complete to be checked.